# Lignin-Based Porous Biomaterials for Medical and Pharmaceutical Applications

**DOI:** 10.3390/biomedicines10040747

**Published:** 2022-03-23

**Authors:** Nan Nan, Wanhe Hu, Jingxin Wang

**Affiliations:** 1Center for Sustainable Biomaterials & Bioenergy, West Virginia University, Morgantown, WV 26506, USA; nnan@mail.wvu.edu (N.N.); wh0030@mix.wvu.edu (W.H.); 2Division of Forestry and Natural Resources, West Virginia University, Morgantown, WV 26506, USA

**Keywords:** lignin, porous biomaterial, nanocomposite, drug delivery, tissue engineering, electrospinning, 3D printing

## Abstract

Over the past decade, lignin-based porous biomaterials have been found to have strong potential applications in the areas of drug delivery, tissue engineering, wound dressing, pharmaceutical excipients, biosensors, and medical devices. Lignin-based porous biomaterials have the addition of lignin obtained from lignocellulosic biomass. Lignin as an aromatic compound is likely to modify the materials’ mechanical properties, thermal properties, antioxidant, antibacterial property, biodegradability, and biocompatibility. The size, shape, and distribution of pores can determine the materials’ porous structure, porosity, surface areas, permeability, porosity, water solubility, and adsorption ability. These features could be suitable for medical applications, especially controlled drug delivery systems, wound dressing, and tissue engineering. In this review, we provide an overview of the current status and future potential of lignin-based porous materials for medical and pharmaceutical uses, focusing on material types, key properties, approaches and techniques of modification and fabrication, and promising medical applications.

## 1. Introduction

Lignin-based porous biomaterials are materials that have the addition of lignin and contain pores. These materials have great potential for application in the medical and pharmaceutical areas of drug delivery, tissue engineering, wound dressing, pharmaceutical excipients, biosensors, medical devices, and more. (Figure 1). Lignin is a class of aromatic biopolymer and a primary component of the cell walls in lignocellulosic biomass, such as woods, grasses, and crop stalks. It is the second most abundant natural polymer after cellulose. The structure of lignin varies depending on extraction processes, biomass species, and growth conditions [1]. The three fundamental monolignols of lignin have been identified, sinapyl alcohol (S), coniferyl alcohol (G), and *p*-coumaryl alcohol (H) (Figure 2). These units are linked repeatedly via chemical bonds, including *β*-O-4, 4-O-5, *β*-5, *β*-1, *β*-*β*, 5-5, spirodienone, dibenzodioxocin, etc. [2]. The different extraction processes can produce different types of lignins. The commonly used lignin extraction processes for medical uses are enzymatic hydrolysis process (hydrolytic lignin) [3,4,5], kraft process (kraft lignin or alkali lignin) [6,7], sulphite process (lignosulfonates) [8,9], and organosolv process (organosolv lignin) [10]. In most cases, further modification of lignin is necessary to enhance the specified properties of lignin-based materials, for example grafting with allyl groups to improve cross-linking [8], and grafting with folic acid to magnetic-functionalize lignin-based nanoparticles [3].

As part of lignin-based biomaterials, including composites, copolymers, nano-/micro-particles and fibers, lignin has been documented as the matrix, crosslinking agent, nano and micro-filler in various studies to build designed structures and/or modify one or more properties of the materials, such as mechanical, thermal, antioxidant, and antibacterial properties, biodegradability, biocompatibility, permeability, porosity, water solubility, and adsorption ability. These materials are most likely to contain pores (Figure 3). The porous structure can influence materials’ geometry, density, surface areas, permeability, and absorption capacity; therefore, it is suitable for some medical uses. For instance, pores can be used to lighten weight in bone tissue engineering applications and applying the absorption ability can be used to control the drug loading and release in drug delivery. In general, the porous structure of the lignin-based materials can be classified into a few categories according to the resource of pores: (1) the initial pores of lignin materials. As a biopolymer, lignin may contain abundant macro-, micro-, and nano-size pores, and the pore’s size and distribution vary with different raw materials and lignin extraction processes. This type of pores can be modified. For example, the pores can be reduced by reducing the lignin particle size using milling or micro-/nano-particle processing equipment. Moreover, thermal, physical, and chemical treatments can change the porosity, such as carbonization, CO_2_, and alkali (e.g., KOH) activations; (2) the pores provided by other polymer(s) of the composite, such as chitosan, cryogel, and cellulose. In most cases, the pore size and distribution depend on the type of polymers and are more uniform; (3) the porosity could be generated or reduced through cross-linking processes among lignin and other polymer molecules [19]; and (4) the porous structure could be designed and fabricated by different methods, for instance mold/die casting (Figure 3a), pressurized CO_2_, and electrospun fibrous mats (Figure 3b).

In this paper, we provide an overview of the significant present study findings on a variety of lignin-based porous biomaterials for medical uses, including hydrogels, nano-/microparticles, fibers, and composites. The fabrication technologies and approaches, materials’ porous structure and critical properties, and corresponding utilizations are described for each type of the prevalent lignin-based materials.

## 2. Lignin-Based Porous Hydrogels

Hydrogels are three-dimensional networks of crosslinked hydrophilic molecules or polymers that can absorb and retain large amounts of water or biological fluids within their structure [1,15,23,24]. Hydrogel products have been developed for a wide range of biomedical applications, including contact lenses, surface lubrication coatings, controlled drug release devices, wound healing dressings, cell immobilization islets, three-dimensional cell culture substrates, and bioactive scaffolds for regenerative medicine [24]. It’s worth mentioning that the swelling capacity of hydrogels as a response to a trigger stimulus could be a highly desirable property to design drug delivery systems, which allows the active agent(s) to be loaded and released under control [1]. Therefore, hydrogel systems for controlled drug delivery, such as biodegradable, smart, and biomimetic hydrogels, have been well studied and considered as promising products for commercial exploitation [25].

Recently, the renewable natural polymers-based hydrogels have attracted greater attention for various applications from manufacturing to biomedical, due to their inherent biocompatibility, biodegradability, and biologically recognizable moieties that support cellular activities, although some of them may have some degree of drawbacks in mechanical properties or unexpected immune responses [24,25]. Hydrogels can be prepared from natural polymers, such as sodium alginate, starch, protein, gelatin, hyaluronate, xyloglucan, hemicelluloses, cellulose, lignin, chitosan, and collagen, and their derivatives [26,27]. The presence of hydrophilic functional groups on the chemical structure of lignin endorses its role as a polymer for hydrogels preparation. Also, lignin’s structure and the presence of rich phenolic and aliphatic groups make this biopolymer capable of more chemical modification and reactions [22]. Lignin has been successfully fabricated into hydrogels combined with other polymers (e.g., chitosan, poly(vinyl alcohol) (PVA), alginate, and cellulose) [23]. Moreover, the attractive properties of lignin, such as biodegradability, biocompatibility, non-toxicity, antioxidant, and antimicrobial properties make it capable of being an excellent component of hydrogels [1,23,28]. Lignin-based hydrogels are more sustainable and environment-friendly than synthetic hydrogels. Therefore, in recent years, lignin-based hydrogels have been increasingly used for many purposes, especially in medical and pharmaceutical areas, such as drug delivery, wound healing, and tissue engineering [23,29].

### 2.1. Porous Structures of Lignin-Based Hydrogels

Pores with different structures and polarities could be formed during the fabrication of lignin-based hydrogels. Porous architecture could provide an extended surface area for cell attachment, and it could enhance cellular ingrowth and vascularization. The developed interconnected porosity was observed for the 3D structure of cross-linked lignin-agarose hydrogel formed by freeze-dry (Figure 3c) [22]. Moreover, the pore sizes, distribution, and homogeneousness could be correlated with the maximum water uptake and impact the swelling capacity, the most critical characteristics of hydrogels. Larrañeta et al. [30] indicated that lignin-poly(ethylene glycol) (PEG) hydrogels showed larger holes and pores in their structure after removal of the water, and a smaller pore size distribution resulted in a lower water uptake. Importantly, studies found that the addition of lignin can increase the porosity (major in macro-pores) of dry hydrogels, such as lignin-gantrez S-97 (poly(methyl vinyl ether) and maleic acid copolymer) (GAN) [30], lignin-PEG [30], and cellulose-lignin [24], at a certain degree to produce a more relaxed network with higher swelling capacity. It is possible to control the drug delivery through increasing lignin content to increase the drug release rate. For instance, Ciolacu et al. [24] found that an increase in lignin content (from 25% to 75%) of cellulose-lignin hydrogels enlarged the average hydrogel pore size from 169 μm to 431 μm and gradually increased the hydrogels’ drug (i.e., polyphenols) release rate from approximately 17% to 29% (Figure 4). Additionally, lignin can lead to a more homogenous and less dense structure of the resulting composite hydrogels [24,26].

### 2.2. Current Status of Lignin-Based Hydrogels for Medical Applications

In the past decade, several clinical research studies have shown promising results for applications of lignin-based hydrogels as materials for medical and pharmaceutical applications, particularly in drug delivery systems, wound healing, and tissue engineering (Table 1).

Drug delivery systems: Lignin-based hydrogels have been used for the controlled drug release of both hydrophobic and hydrophilic compounds due to lignin containing both hydrophobic and hydrophilic groups naturally. For instance, the release of hydrophobic curcumin is used for cancer treatments. The hydrogels (i.e., lignin-GAN and lignin-PEG) sustained the delivery of this compound for up to 4 days [30]. The release of hydrophilic bisoprolol fumarate is for high blood pressure and heart failure treatments. Epichlorohydrin (ECH) as a cross-linking agent with crosslinked xanthan and lignin epoxy-modified resin mixture was the superabsorbent hydrogel with high swelling rate in aqueous mediums and loaded 14.4–19.2% bisoprolol fumarate drug in polymer matrix [31].

Wound dressing and tissue engineering: Carrageenan–lignin–silver nanoparticles (AgNPs)-MgCl_2_ hydrogels, adding AgNPs capped by lignin to the carrageenan matrix cross-linked with MgCl_2_ divalent cations, significantly healed the wounds in Sprague-Dawley rats within two weeks, reducing the wound area to lower than 3% [32]. A study found that the introduction of lignin effectively improved the mechanical strength (tensile stress was up to 46.87 MPa), protein adsorption capacity, and wound environmental regulation ability of the chitosan-PVA hydrogel, since its mechanical properties are poor and it cannot satisfy the requirements of wound dressing as an environmental conditioner to accelerate wound healing. The lignin–chitosan–PVA composite hydrogel significantly accelerated wound healing in a murine wound model [33]. Eivazzadeh-Keihan et al. reported that lignin in a three-dimensional hydrogel network significantly enhanced the mechanical properties of agarose, a neutral and biocompatible polysaccharide with very good self-gelling properties, which can generate thermo-reversible hydrogels [22]. The wounds of mice treated with the cross-linked lignin–agarose–silk fibroin (SF)-ZnCr_2_O_4_ nanobiocomposite scaffold were almost completely healed in five days in in vivo assay experiments. The enhanced mechanical properties and elastic network of this nano-biocomposite mean that it has potential for tissue engineering. Ravishankar et al. indicated that alkali lignin crosslinked chitosan hydrogels were non-toxic to Mesenchymal stem cells and to zebrafish up to 100 μg/mL in in vivo experiments [34]. NIH 3T3 mouse fibroblast cells showed good cell migration characteristics, suggesting that the hydrogel might be suitable for wound healing. Furthermore, the lignin–chitosan hydrogels provided a conducive surface for cell attachment and proliferation, making it suitable for application as scaffolds in tissue engineering. The developed lignin-based hydrogels provide new opportunities for highly efficient skin wound care and management.

**Table 1 biomedicines-10-00747-t001:** Literature overview of lignin-based hydrogels and their applications.

Hydrogel	Methodology	Advanced Properties	Application	Ref.
lignin-agarose-SF- ZnCr_2_O_4_	cross-linking reaction between lignin and agarose by ECH, and addition of SF and ZnCr_2_O_4_	cellular ingrowth & vascularization in designed scaffolds	wound healing, tissue engineering	[22]
carrageenan-lignin-AgNPs-MgCl_2_	one-pot synthesis of AgNPs using lignin as a reducing and capping agent in the carrageenan matrix cross-linked with divalent cations	wound healing speed	wound dressing	[32]
chitosan-alkali lignin	combining lignin with an aqueous-acidic solution of chitosan	forming electrostatic cross-links between the chitosan chains	wound healing, tissue engineering	[34]
lignin-chitosan-PVA	mixing an aqueous acidicsolution of chitosan with solutions of lignin and PVA	mechanical strength, protein adsorption, wound environmental regulation ability	wound dressing	[33]
lignin-GAN; lignin-PEG	esterification reaction with microwave radiation	welling, sustaining drug delivery, adhesion reduction	drug delivery (curcumin)	[30]
xanthan-lignin	mixing lignin with xanthan using ECH as crosslinking agent	the amount of drug loading	drug delivery (bisoprolol fumarate)	[31]
cellulose-lignin	mixing cellulose alkaline solution with lignin, followed by the crosslinking with ECH	the swelling capacity and drug release	drug delivery (polyphenols)	[24]

## 3. Lignin-Based Hollow Nanoparticles

Studies found that the hollow particles can be widely applied in bioimaging, diagnostics, and drug delivery applications [35,36]. Lignin is a promising raw material for the preparation of nanomaterials in medical and biological applications due to its biodegradability, non-cytotoxicity, and abundant availability [4]. Significantly, lignin nanoparticles have shown superior properties in antibacterial, antioxidation, and UV barrier [23,37,38]. Lignin-based hollow nanoparticles (LHNPs) have been considered to be suitable for drug and gene delivery systems [7,23,39].

### 3.1. Hollow Structure of the LHNPs

Hollow sphere particles consist of two parts, an external shell and an internal void (Figure 5). The shell may be formed by one (single-shelled) (Figure 5a) or several (multi-shelled) walls/layers (Figure 5b). Furthermore, the thickness, permeability, morphology, pore size distribution, and number of wall layers are significant factors for different applications. For instance, multi-shelled hollow particles may have complex interiors with multiple cages, tunnels, and pores. Their advanced structure enables a higher density of the structure and better entrapment of the active compound in the particle interior, which is important for the undesired and premature release of entrapped active compounds. Ideal hollow spheres can exhibit excellent loading capacities and porous structure, as well as a certain degree of mechanical, chemical, and thermal stability, integrity, and sensitivity [36].

Zhou et al. studied the LHNPs with multiple shell walls for the hydroxycamptothecin (HCPT) delivery, an antitumor drug [4]. The SEM image (Figure 6a) of LHNPs shows the spherical hollow structure with a single hole on the surface of the particles. The transmission electron microscopy (TEM) image (Figure 6b) displays a clear contrast between the center and the shell, which further supported the hollowness of the particles. The study by Zhou et al. showed that LHNPs have efficient encapsulation and good capacity for sustained-release medication delivery (i.e., the antitumor drug), and the *β*-cyclodextrin (*β*-CD) grafted LHNPs exhibited a better encapsulation capability because their higher specific surface area and the porosity values.

### 3.2. Current Status of Lignin Nanoparticles for Pharmaceutical and Medical Applications

Lignin nanoparticles have been demonstrated as capable of loading drugs (Table 2) for targeted cancer and tumor treatments, and the developed drug delivery nanosystems enhanced the uptake of nanoparticles by cells [23]. Both hydrophobic and hydrophilic compounds have been successfully loaded on many types of lignin nanoparticles. For example, the reported high loading capacity of hydrophobic molecules or poorly soluble drugs are curcumin [10], hydrophobic coumarin-6 [8], bioactive molecule resveratrol [40], hexadecane [6], sorafenib (SFN), benzazulene (BZL), capecitabine (CAP) [7], and HCPT [4]. The studied high loading capacity of hydrophilic molecules are rhodamine 6G [9] and doxorubicin hydrochloride (DOX) [3]. Moreover, different types of lignin have been used to produce nanoparticles, such as enzymatic hydrolysis lignin [3,4], kraft/alkali lignin [6,7,39,40], lignosulfonate [8,9], and organosolv lignin [10]. The lignin is usually modified to improve the network structure of lignin molecules to increase specific surface area and porosity, then the hollow nanoparticles via self-assembly to encapsulate and load the drugs; for example, the *β*-CD modified LHNPs exhibited a good sustained-release capability to the antitumor drug HCPT [4]. Besides the specific interaction between biological drug-loading materials and drugs, other important parameters for drug delivery systems include the temperature and pH at which the entrapped drug is released to the selected sites [36].

## 4. Lignin-Based Porous Biocomposites

Studies found that lignin-based biocomposites have great potential to be applied in drug delivery systems, wound dressing, tissue engineering, and regenerative medicine (e.g., skin, nerve, cartilage, bone, and hard tissues) (Table 3). Generally, the lignin has been used to enhance various biocomposites in mechanical properties, porosity for material density and controlled drug release, and antibacterial and antioxidant properties. For instance, lignin–chitosan biocomposites [41], lignin–alginate cryogels [42], and lignin–gelatin cryogels [43] fabricated through a freezing technique have been studied for wound healing and tissue repairing. The unique freezing temperatures allow ice crystals to grow, and rapid freezing with subsequent freeze-drying led to porous structures [42]. Memic and Abudula found that lignin improved the mechanical performance of the cryogel and enhanced its shape recovery rate. The microporous structure exhibited excellent free radical scavenging activity and inhibited the growth of both gram-positive and -negative bacteria [43]. Alginate–lignin aerogels are studied for tissue engineering and regenerative medicine [44]. In this study, lignin was used to reduce the hydrophilicity of alginate and hence provide a more suitable environment for cells to adhere, grow and differentiate, and abate the scaffold degradation rate to match with the rate of new bone tissue regeneration. Moreover, the foaming procedure by rapid expansion of CO_2_ introduced macroporosity into the aerogels. Chitin nanofibrils and nanolignin complexes loaded with glycyrrhetinic acid (GA) are studied for skin regeneration [45].

Erakovic et al. conducted several studies of hydroxyapatite (HAP)–lignin and Ag-HAP–lignin coatings on titanium by electrophoretic deposition (EPD) for tissue implant or repairing coatings [46,47,48,49,50]. Their results showed that higher lignin concentrations protect the HAP lattice during sintering and improve the corrosion stability of the HAP–lignin and Ag–HAP–lignin coatings [46,50]. Lignin also accommodated a better interconnected porous structure and modified the surface porosity that enables osteogenesis. Ag-HAP–lignin coating showed higher reduction of bacteria Staphylococcus aureus TL than did HAP-lignin coating. Cytotoxicity assay revealed that both coatings can be classified as non-toxic against healthy immunocompetent peripheral blood mononuclear cells (PBMC). Furthermore, there are plenty of studies of lignin-based micro- or nano-fibers and composites and the further generated composite/fibrous membranes and scaffolds for tissue engineering (i.e., to support cells and generate extracellular matrix) [5,51,52,53,54,55,56].

Lignin can be used as a precursor to make porous carbon materials through thermochemical approaches, such as lignin-based carbon for controlled drug delivery [57]. Lignin-derived natural adhesives can be applied in adhesive-consuming materials in medical applications, such as particleboard as a tissue-substitute phantom material [58]. Additionally, lignin can be used as a pharmaceutical excipient to improve the bioavailability of drugs, which can be an important factor in oral dosage development, and to enhance the drug (e.g., aspirin) release rate from tablets [59,60]. The aquasolv lignin is safe to consume and presents antioxidant and antidiabetic capacity, properties that can add to solid dosage forms in pharmaceuticals [61]. Lignin also can be used as an antioxidant [16,23,62,63], such as to resist oxidation and ultraviolet radiation (UV) and increase the sun protection factor (SPF) of sunscreen products [23].

## 5. Promising Fabrication Technologies of Lignin-Based Medical Materials

Unfortunately, the lignin-based medical materials are still in the lab research stage. Like most commercialized higher-value materials, a more feasible, highly-efficient, mature, and cost-effective technology is more likely to be commonly used, promoted, and commercialized. According to present research, the promising techniques are electrospinning and 3D printing.

### 5.1. Electrospinning

Electrospinning, as one of the most versatile manufacturing technologies in the past decade, has been used to produce lignin-based composites for biomedical applications due to its great diversity of fabricating nanofibers/nanofibrous scaffolds featuring high aspect ratio, large specific surface area, high porosity, flexibility, structural abundance, surface functionality, and capability to deliver bioactive agents [23,64,65]. Electrospinning is a method to produce ultrafine fibers by charging and ejecting a polymer melt or solution through a spinneret under a high-voltage electric field and includes the type of blend, coaxial, emulsion, and needleless (Figure 7) [64,66,67].

Coaxial electrospinning has been used by Abudula et al. to encapsulate the chitin-lignin gels with polycaprolactone (PCL) (Figure 8a), and the fabricated gel fibrous scaffolds showed controlled drug release ability [5]. Abudula et al. fabricated hybrid nanofibrous scaffolds composed of a chitin–lignin sol–gel and solution of elastomeric poly(glycerol sebacate) (PGS) using a blend electrospinning technique [52]. Salami et al. used electrospun PCL-lignin scaffold to enhance the biological response of the cells with the mechanical signals (Figure 8b) [53]. Their results showed that the addition of lignin (10 wt.%) improved the porosity, biodegradation, minimum fiber diameter, optimum pore size, tensile strength, and young modulus. Saudi et al. used lignin-filled PVA-PGS fibers for nerve tissue engineering and found that the addition of lignin (1 wt.%–5 wt.%) reduced the fiber diameter, increased the modulus of elasticity, and enhanced cell proliferation [54]. Poly(lactic acid) (PLA)-lignin and poly(L-lactide) (PLLA)-lignin nanofibrous membranes have been used to repair superficial articular cartilage defects after implantation [55]. Lee et al. fabricated lignin-decorated thin multi-walled carbon nanotubes (*t*-MWNTs)-PVA nanofibrous webs, and found that the webs have many small pores among very thin nanofiber; these pores are likely to protect the human body against external impacts while allowing the diffusion of water vapor that is generated by the human body [56].

### 5.2. Three-Dimensional Printing

Three-dimensional (3D) printing is an additive manufacturing process for constructing 3D physical objects from a computer-aided design (CAD) model through the successive layer-by-layer deposition of materials, such as powders, epoxy resins, thermal plastics, and certain gel-like biomaterials [68,69,70]. This technology is promising for medical and pharmaceutical applications due to the ability to fabricate patient specific and custom-made medical products, equipment, and drugs. Lignin-based materials are very likely to be processed by 3D printing, since some of them have potential to meet both printing and medical requirements, such as appropriate printability, mechanical properties, biodegradability, biocompatibility, tissue biomimicry, and non-cytotoxicity [69,70,71,72]. For instance, Domínguez-Robles et al. used 3D printing for PLA–lignin to design meshes for wound dressing and fabricate capsules for pharmaceutical oral drug carriers (Figure 9) [16].

Furthermore, the 3D printing approaches, including the choice of printing technologies (e.g., stereolithography (SLA), fused deposition modeling (FDM), direct ink writing (DIW), and bioprinting) and the formulation of lignin-based inks, remain to be further developed. SLA and FDM methods are commonly used in polymer 3D printing, such as parts, anatomical models, and microfluidics, which are most likely be applied to lignin-based materials due to the polymer nature of lignin and mixed component(s). FDM is quicker and more cost-effective but generates relatively rough surface finishes and lacks strength. Therefore, the thermoplastic type of lignin-based materials are likely suitable inks for FDM and SLA printing. For instance, methacrylate resins filled with lignin-coated cellulose nanocrystals have been used as inks for SLA printing [73,74]. Moreover, the lignin-based hydrogels are most likely to be good as innovative hydrogel bioinks for 3D bioprinting in tissue engineering, including inkjet, microextrusion, and laser-assisted bioprinting techniques. Inkjet bioprinting dispenses picoliter droplets of the bioink onto a substratum through a non-contact process; a microextrusion printer ejects droplets and can move stages along three XYZ orthogonal planes; and a laser-assisted bioprinter uses the optical properties of the bioink or the wavelength of the laser to direct ink ejection [75]. Additionally, there are still many challenges to apply 3D printing technology widely in the medical field. These challenges include the lack of regulations concerning the manufacture of 3D-printed products and the fact that most of the bioprinting products (e.g., various tissues and organs) are at the laboratory level and may require more time and effort to achieve the next level of clinical and commercial application. Nonetheless, this technology has introduced significant advantages and infinite possibilities for medical and pharmaceutical applications.

## 6. Conclusions

In this review, we have shown that the majority of studies on lignin-based porous biomaterials for medical and pharmaceutical uses, such as drug delivery systems, wound dressing and tissue engineering, have shown great potential for development and applications. Although the related research is still rather limited, lignin has strong potential to make a variety of medical products, including hydrogels, micro-/nanoparticles, fibers, capsules, scaffolds, fibrous mats, members, webs, etc., due to its advantages in biodegradability, biocompatibility, and antioxidant, antibacterial, thermal, and mechanical properties. Nevertheless, there are challenges that restrict the development of lignin-based medical materials, for instance the complex and inhomogeneous feature of lignin, technical limitations of synthetic processes, and legal and regulatory issues for the applications of medical materials. It is anticipated that advances in technology will further help solve problems in lignin-based biomaterials and design more high-value products with the desired shape, size, and performance.

## Figures and Tables

**Figure 1 biomedicines-10-00747-f001:**
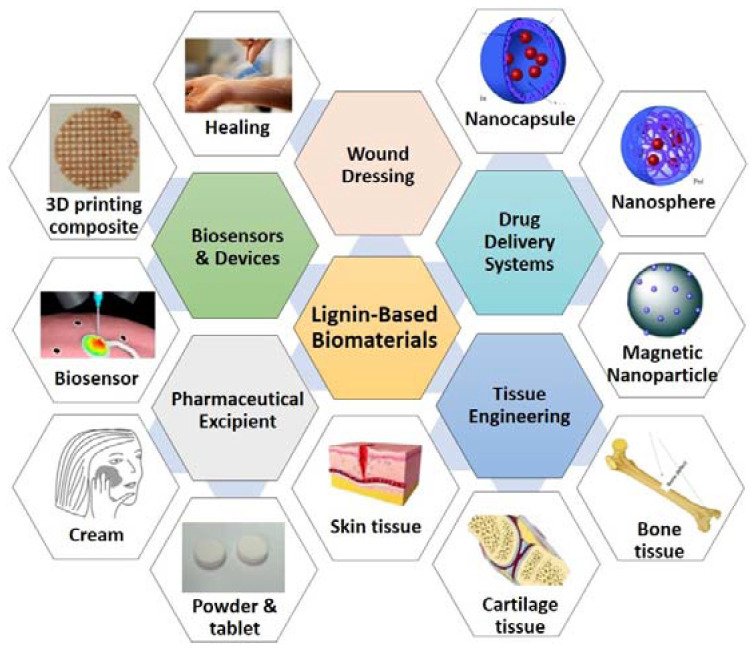
Schematic illustration of the medical applications of lignin-based biomaterials. (Original images’ resources: nanocapsule and nanosphere [11], magnetic nanoparticle [12], skin tissue and cartilage tissue [13], bone tissue [14], wound healing and biosensor [15], 3D printing composite [16], powder and tablet [17], and cream [18]).

**Figure 2 biomedicines-10-00747-f002:**
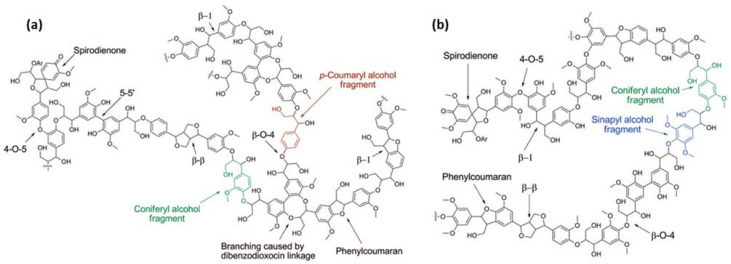
Schematic representation of (**a**) a softwood lignin structure consisting of repeated H (highlighted in red) and G (highlighted in green) unites and (**b**) a hardwood lignin structure consisting of repeated S (highlighted in blue) and G units (reproduced from Ref. [2] with permission from the American Chemical Society).

**Figure 3 biomedicines-10-00747-f003:**
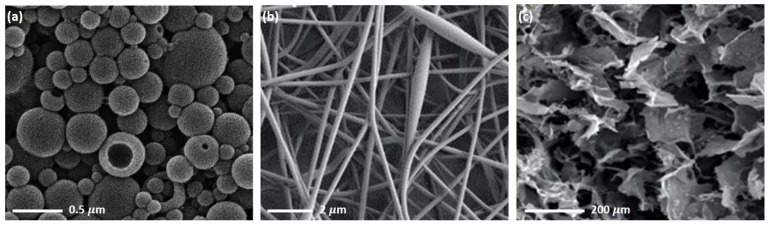
Porous structures in the scanning electron microscopy (SEM) images of (**a**) lignin-based nanocapsules (reproduced from Ref. [20] with permission from the American Chemical Society), (**b**) lignin-based fiber mat (reproduced from Ref. [21] with permission from the Royal Society of Chemistry), and (**c**) lignin-based hydrogel (reproduced from Ref. [22] with permission from the Royal Society of Chemistry).

**Figure 4 biomedicines-10-00747-f004:**
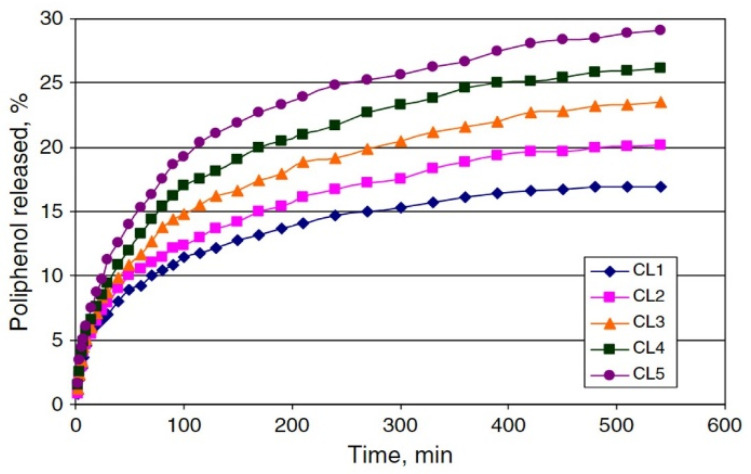
Release profiles of polyphenols from cellulose-lignin (CL) hydrogels in water:ethanol medium, at 37 °C. The lignin contents from CL1 to CL5 are 25%, 33%, 50%, 67%, and 75%, and the average pore sizes are 203 μm, 215 μm, 233 μm, 307 μm, and 431 μm, respectively (reproduced from Ref. [24] with permission from Elsevier).

**Figure 5 biomedicines-10-00747-f005:**
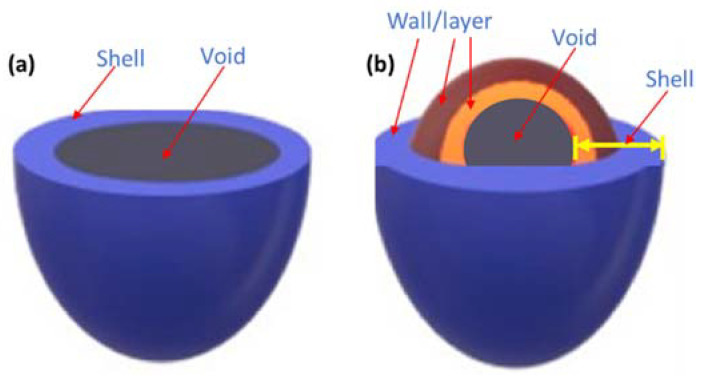
Types of hollow spherical particles: (**a**) single-shelled structure and (**b**) multi-shelled structure (reproduced from Ref. [36] with permission from the Elsevier).

**Figure 6 biomedicines-10-00747-f006:**
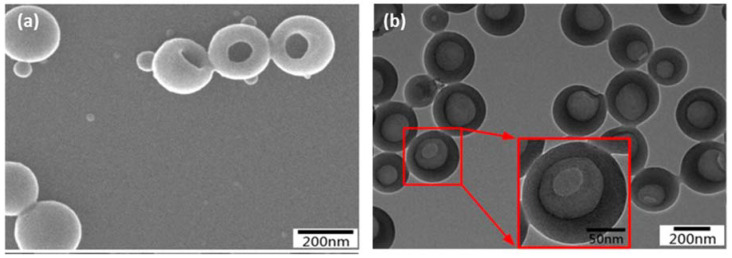
Image of LHNPs (**a**) SEM and (**b**) TEM (two red arrows point to an enlarged image of a selected particle) (reproduced from Ref. [4] with permission from the Elsevier).

**Figure 7 biomedicines-10-00747-f007:**
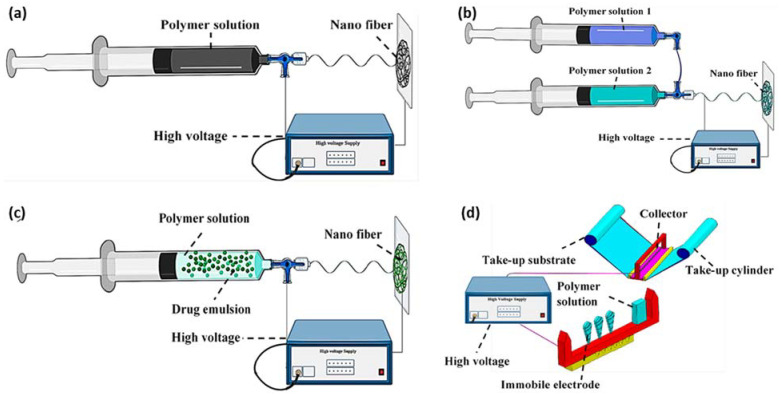
Electrospinning technologies (**a**) blend electrospinning, (**b**) coaxial electrospinning, (**c**) emulsion electrospinning, and (**d**) needleless electrospinning (reproduced from Ref. [67] with permission from the MDPI).

**Figure 8 biomedicines-10-00747-f008:**
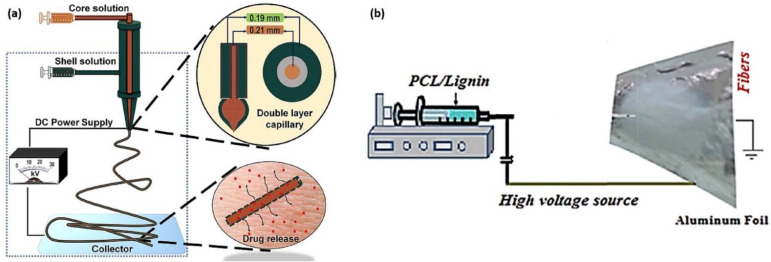
Examples of (**a**) the coaxial electrospinning of chitin–lignin-based hybrid fiber encapsulation by PCL (reproduced from Ref. [5] with permission from Springer Nature), and (**b**) the blend electrospinning of PCL–lignin nanocomposite scaffold (reproduced from Ref. [53] with permission from the PMC).

**Figure 9 biomedicines-10-00747-f009:**
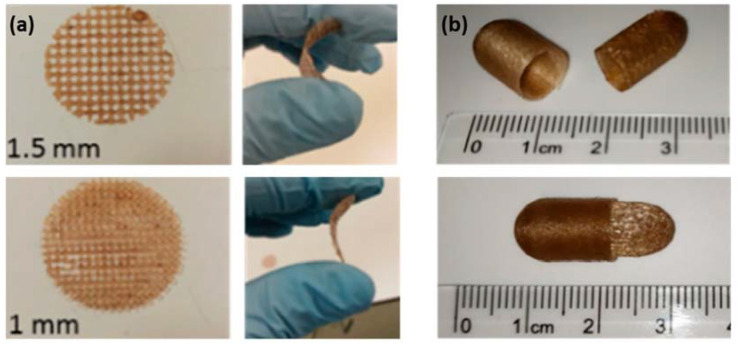
3D printed (**a**) meshes and (**b**) oral capsules (reproduced from Ref. [16] with permission from the MDPI).

**Table 2 biomedicines-10-00747-t002:** Literature overview of lignin-based nanoparticles (LNPs) and their applications in drug delivery.

Lignin	Methodology	Advanced LNPs	Drug	Ref.
enzymatic hydrolysis lignin	LNPs covered by Fe_3_O_4_ nanoparticles and grafted with folic acid; lignin was dissolved in THF and water for self-assembly	folic-magnetic-functionalized LHNPs	DOX	[3]
enzymatic-hydrolysis lignin	grafting *β*-CD onto lignin; modified lignin prepared LHNPs via self-assembly to encapsulate HCPT.	LHNPs and *β*-CD-LHNPs	antitumor drug HCPT	[4]
kraft lignin	methacrylation of lignin, mini-emulsion with solvent evaporation	lignin nanocarriers	hexadecane	[6]
kraft lignin (softwood)	solvent exchange by dissolving lignin in tetrahydrofuran (THF) and adding water into the system via dialysis	LNPs, magnetic Fe_3_O_4_-LNPs	SFN, BZL, CAP	[7]
Sodium lignosulfonate	grafting lignin with allyl groups by etherification, further dispersed in an oil-in-water miniemulsion system by ultrasonication, then form nanocapsules via a thiolene radical reaction	lignin-based pH-responsive nanocapsules	coumarin-6	[8]
low-sulfonated lignin	anti-solvent preparation: lignin was dissolved in ethylene glycol and precipitated by HCl	LNPs	rhodamine 6G	[9]
organosolv lignin	using a modified phase separation	LNPs	curcumin	[10]
alkali lignin (sorghum, loblolly pine, poplar, sugar cane bagasse)	synthesized with an alumina membrane template	lignin nanotubes	DNA	[39]
alkaline lignin	diluting the stock solution with organic solvent, then self-assembly	magnetic Fe_3_O_4_-LNPs	resveratrol	[40]

**Table 3 biomedicines-10-00747-t003:** Literature overview of lignin-based biocomposites, technologies, and applications.

Lignin	Lignin-Based Biocomposites	Technique	Application	Improved Properties	Ref.
lignin (*artocarpus heterophyllus* waste)	lignin-chitosan biocomposites	freezing technique	wound healing & dressing	mechanical stability	[41]
enzyme hydrolysis lignin (wheat straw)	lignin-alginate cryogels porous scaffolds	freezing technique	tissue engineering, regenerative medicine	porous structure	[42]
lignin nanoparticles	lignin-gelatin cryogels	freezing technique	wound healing, tissue engineering	mechanical properties, antibacterial	[43]
alginate-lignin (wheat straw)	wet and dry alginate-lignin aerogels	using CO_2_ induced gelation	tissue engineering, regenerative medicine	reduce hydrophility of alginate	[44]
bio-lignin	chitin nanofibrils and nanolignin complexes loaded with GA	spray drier	skin regeneration	cytocmpatible, anti-inflammatory	[45]
organosolv lignin (North American hardwoods)	HAP-lignin and Ag-HAP-lignin coatings on titanium	modified chemical precipitation, and EPD	bone and hard tissue implant/repairing coatings	Surface porosity, antibacterial	[46,47,48,49,50]
alkali lignin	lignin-chitosan microfibres	wet-spinning	tissue repair & regeneration	mechnical properties	[51]
bio-lignin	PLC-chitin-lignin gel fibrous scaffolds	coaxial electrospinning	drug release, wound dressing	controlled drug release, antibacterial	[5]
bio-lignin	chitin-lignin sol-gel nanofibrous scaffolds	electrospinning	wound care products	mechanical properties, antibacteria	[52]
lignin (Mn: 3000; Aldrich)	PCL-lignin nano scaffold	electrospinning	tissue engineering	biological response of the cells	[53]
lignin	nanofibrous PVA-PGS-lignin scaffolds	aligned electrospinning	nerve tissue engineering	neural cell proliferation, differentiation	[54]
lignin	lignin-PLA, PLLA-lignin nanofibrous membranes	electrostatic spinning	cartilage tissue engineering	mechanical properties	[55]
lignin (Sigma Aldrich, Korea)	lignin-*t*-MWNT-PVA nanofibers	electrospinning	wound healing	antimicrobial properties	[56]
kraft lignin (softwood)	lignin-PLA nanocomposite	3D printing	wound dressing	mechanical & surface properties, antioxidant	[16]

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
