# Peer review of "Lignin-Based Porous Biomaterials for Medical and Pharmaceutical Applications"

_biomedicines, 2022, doi:10.3390/biomedicines10040747_

Round 1

Reviewer 1 Report

A well written review article. The subtitles are cathergorized in a good way and the tables present the information well summarised. The authors have performed good research on the field of the topic with relevant references. The authors could mention more about the disadvantages for the materials in the different sections where mostly the pros but no cons are mentioned. However, it is summarised in the conlusion.

See the manuscript with minor comments.

Author Response

Comment 1: Figure 3. Please make the scale bars in the figure similar.

Response 1: We have revised all the scale bars in Figure 3 to make them uniformed (see Figure 3 in line 77 on page 3).

Comment 2: Repeating problem in lines 129-133.

Response 2: In the revised manuscript, the modification was made (see line 132-136 on page 4)

Comment 3: Spelling mistakes in 3 places.

Response 3: In the revised manuscripts, the spelling mistakes were corrected (see track changes in lines 28, 276, and 317).

Reviewer 2 Report

Comments to the Author:

Chemical names and formulas carefully checked. The same style for structure is favorable.

a few aspects, which could be improved upon to consolidate the manuscript. The manuscript should be reconsidered after correction. I suggest the following modifications:

Author described different lignin-based porous materials for biomedical application. It is desirable to have some discussions on difference between and advantage and disadvantage of each type. This manuscript is a review article. Some sections in the text seems unconnected and separate from each other. So, the brief discussion at the end of each section is desirable.

In the paragraph 5 author considering electrospinning and 3D printing as the promising fabrication techniques. I suggest that the authors should add some arguments and discussion related their choice.

line 227-229: Sentence should be reconsidered and rewritten.

Chemical names and formulas should be carefully checked. The same style for structure is favorable.

Author Response

Comment 1: Author described different lignin-based porous materials for biomedical application. It is desirable to have some discussions on difference between and advantage and disadvantage of each type. This manuscript is a review article. Some sections in the text seems unconnected and separate from each other. So, the brief discussion at the end of each section is desirable.

Response 1: Section 4 and Table 3 was revised (see track changes on page 10).

The first paragraph of both sections 2 and 3, and the first couple of sentences of section 4 briefly discussed the advantages of lignin-based hydrogels, nanoparticles, and biocomposites, respectively. The disadvantages were discussion associated with illustrated findings of each individual case/study, since the studies at the current stage are very limited, and the focus and fabrication method of these studies were rather varied. Also, the Conclusions section discussed the disadvantages.

Comment 2: In the paragraph 5 author considering electrospinning and 3D printing as the promising fabrication techniques. I suggest that the authors should add some arguments and discussion related their choice.

Response 2: Section 5 was revised according to the reviewer’s comments (see track changes on page 14)

Comment 3: line 227-229: Sentence should be reconsidered and rewritten.

Response 3: This sentence was revised according to the reviewer’s comments (see track changes in lines 235-241 on page 9)

Comment 4: Chemical names and formulas should be carefully checked. The same style for structure is favorable.

Response 4: The chemical names, formulas, and abbreviations were checked and revised as needed (see track changes in the revised manuscript).

Reviewer 3 Report

The review describes lignin-based porous materials for medical and pharmaceutical uses, focusing on material types, key properties, approaches and techniques of modification and fabrication, and promising medical applications. The review is well written and describing lignin based hydrogels, nanoparticles, porous bicomoposites for various biomedical applications. I have following concern:

  • The figures have been taken from previous published papers. Have the Authors taken copyright permission for that? If yes, the ligand should be modified Reproduced from Ref [] with permission.
  • The tables in the manuscript should have an additional column describing the major outcomes of the study.

Author Response

Comment 1: The figures have been taken from previous published papers. Have the Authors taken copyright permission for that? If yes, the ligand should be modified Reproduced from Ref [] with permission.

Response 1: Figure 3 was revised, and the old Figure 4 was deleted because of the permission issue. “reproduced from Ref. [#] with permission” was added for each figure. (see track changes in the revised manuscript).

Comment 2: The tables in the manuscript should have an additional column describing the major outcomes of the study.

Comment 2: Table 3 was revised with adding an additional column according to reviewer’s comments (see track changes in table 3 at page 11)

Tables 1 and 2 have the major outcomes column, that is, the advanced properties column in Table 1 and the advanced LNPs column in Table 2.

Round 2

Reviewer 2 Report

Authors done a certain corrections. Nevertheless all chemical formulas and structures should be checked and corrected. Some formulas were corrected, some not (see Table 1 and Table 2).

Author Response

Comment 1: Authors done a certain corrections. Nevertheless all chemical formulas and structures should be checked and corrected. Some formulas were corrected, some not (see Table 1 and Table 2).

Response 1: The chemical formulas and structures were checked and revised as needed (please see track changes in the revised manuscript).